# Iron-borane catalyzed carbonyl hydroboration and isolation of an iron(I)-ketyl radical

Laura A. Grose[1], Ryan J. Schwamm[1], Adam Brookfield[1], David Robinson[2] ✉ & Darren Willcox[1] ✉

Hydroboration of carbonyl compounds has proven a pivotal route to access alcohols and other C1 moieties in recent years. Despite this, iron-based catalyst systems are somewhat limited with very little mechanistic understanding of these systems developed. Here we show that an iron metalloborane complex [{($^{iPr}$DPB$^{Ph}$)Fe}$_2$(μ−1,2-N$_2$)] (**A**) is an efficient pre-catalyst for hydroboration of ketones, cyclic esters and CO$_2$ with mild conditions. Mechanistic insights reveal a previously unreported direct iron(0)-mediated ligand-to-ligand hydride transfer (LLHT) process is in operation with B−H bond breaking being rate determining, indicating the importance of mechanistic studies on well-known transformations. An iron(I)-benzophenone ketyl radical with a unique $S = 1$ antiferromagnetic ground state has been isolated and fully characterized.

Alcohols are privileged structural motifs, which have found widespread application in the pharmaceutical, cosmetic, fragrance and agrochemical industries[1–3]. To date, the industrial route to simple alcohols involves the reduction of a carbonyl compound in the presence of either hydrogen gas at elevated pressure or the use of other alcohols in transfer hydrogenation processes[4–7]. Beyond ketones and other simple carbonyl-containing molecules, carbon dioxide can also be efficiently reduced to C$_1$ hydrocarbons, formaldehyde, formic acid and methanol. These products serve as valuable building blocks for the synthesis of fine and commodity chemicals. While traditionally, hydrogenation of CO$_2$ remains the most atom-efficient route, it necessitates harsh reaction conditions. In the search for milder conditions, homogeneous catalytic hydroboration represents a valuable strategy for the synthesis of alcohols. This two-step process proceeds via the addition of a B−H bond to a carbonyl group to generate a boryl ether that, after hydrolysis, furnishes the desired alcohol[8–11].

Catalysts derived from s-block[12–19], transition metals[20–29], p-block[30–39], and f-block[40–44] element complexes have been employed in this transformative process. However, reports that employ iron-based catalysts are somewhat limited.

In 2017, the group of Findlater reported the first room temperature hydroboration of carbonyl compounds with 10 mol% of Fe(acac)$_3$ in the presence of NaHBEt$_3$ as an activator[45]. The group of Zhang demonstrated the hydroboration of aldehydes and ketones facilitated by an iron(II) coordination polymer, under aerobic conditions[28]. The Fe$_2$O$_3$-nanoparticle catalyzed hydroboration of ketones and aldehydes was reported by Geetharani and Bose in 2018[25]. Expanding beyond ketones, there has been more exploration into utilizing CO$_2$ with iron catalysts. Bontemps and Sabo-Etienne demonstrated that Fe(H)$_2$(dmpe)$_2$ could hydroborate CO$_2$ with various borane agents, leading to the formation of methoxyboranes or bis(boryl)acetals[46]. Moreover, Bontemps combined Fe-catalyzed bis(boryl)acetal production with organic and chemoenzymatic catalysis in a one-pot reaction, achieving enantioselective transformation of CO$_2$ into carbohydrates[47].

Cantat and Berthet reported an iron complex supported by a tripodal phosphine ligand PhSi(XPPh$_2$)$_3$, where (X = CH$_2$, O) for the reduction of CO$_2$ to methoxyborane with 9-Borabicyclo[3.3.1]nonane dimer (9-BBN$_2$) at 60 °C[48]. Webster contributed to the field by utilizing an air-stable [Fe(salen)]$_2$-μ-oxo pre-catalyst with HBpin for the reduction of ketones, aldehydes, and CO$_2$[49]. More recently, Wang and co-workers have demonstrated cooperative catalysis with Cp*Fe-(Cy$_2$PN = C$_5$H$_4$N), showcasing sequential CO$_2$ hydroboration and N-formylation[50]. To the best of our knowledge, at the time of print, there has been no reported iron-catalyzed double hydroboration of cyclic esters.

[1]Department of Chemistry, University of Manchester, Oxford Road, Manchester, UK. [2]Department of Chemistry and Forensics, School of Science and Technology, Nottingham Trent University, Clifton Lane, Nottingham, UK. ✉e-mail: david.robinson@ntu.ac.uk; Darren.willcox@manchester.ac.uk

**Fig. 1 | This work on the hydroboration of carbonyl compounds catalyzed by pre-catalyst complex A.** Top left: ketones; top right: lactams, and bottom left: carbon dioxide.

## Results and discussion

We have previously demonstrated that a diphosphinoborane-iron complex (complex **A**, Fig. 1) is efficient for the catalytic hydroboration of olefins[51]. Building upon these findings, we sought to expand the catalytic capabilities of **A** by exploring its potential in C−O multiple bond reductions, including ketones, cyclic esters, and $CO_2$, using HBpin, and in the case of $CO_2$, (9-BBN)$_2$. Additionally, we aimed to gather kinetic information on the hydroboration of ketones to gain insights into the catalytic cycle.

### Results and discussion

To begin our study, we examined whether **A** would be an efficient pre-catalyst for hydroboration of ketones. We conducted a series of optimization reactions, using our model substrate, cyclohexylphenyl ketone and the economical reductant, HBpin (see ESI for optimization table). A preference for non-coordinating aryl solvents was observed, namely benzene and toluene, with benzene returning a slightly higher yield. Under optimized conditions, the reaction proceeded at ambient temperature with 1 mol% catalyst loading of **A**, resulting in an excellent isolated product yield within 5 h (99%). We employed a control reaction with catalyst-free conditions, after 96 h at ambient temperature, the isolated boronic ester was obtained in a significantly lower yield of 23%.

With the optimal conditions in hand, we turned our attention to exploring the scope of ketones (Fig. 2). Simple aromatic ketones such as acetophenone (**2a**) and its derivatives all underwent clean conversion in less than 10 min, exhibiting a resilience to electron-donating/-withdrawing substituents and steric bulk around the aromatic ring on the phenyl ring (**2b**−**e**). Ketones bearing cyclobutyl or thiophene groups (**2f** and **2g**, respectively) were also successfully hydroborated in less than 10 min with > 95% isolated yield. When the steric bulk of the aliphatic group was increased from a methyl to a tert-butyl (**2h**) or cyclohexyl (**2i**) group, increased reaction times were required, with a moderate decrease in yield for **2h** also being observed. The limitation of steric bulk on the performance of pre-catalyst **A** became evident in **2m,** where, after 24 h at ambient temperature, the hydroborated product was achieved in a moderate 61% yield, while the less encumbered **2l** was obtained in 91% yield after 30 min. This discrepancy suggests that the added methyl groups on **2m** are enough to hinder the transfer of the hydride. As we have previously reported, complex **A** cannot tolerate simple halogen substituents (Cl, Br and I), with the exception of fluorine, due to irreversible degradation of the iron center[52]. Furthermore, hydroboration of a pyridine-containing ketone **2k** resulted in the selective reduction of the ketone without any observed reduction of the pyridine moiety. Notably, when cyclohexen-2-one was used as a substrate, complete 1,2-selectivity was observed (**2n**) with the olefin moiety remaining intact after 24 h.

With these encouraging results, we extended our investigation to other C=O-containing systems. Based on the exquisite chemoselectivity observed in product **2r**, we wondered if ester-containing substrates such as methyl benzoate derivatives could be reduced; however, even with high catalyst loading (20 mol%) and elevated temperatures (110 °C), this was not possible with only starting material being recovered. This selectivity can be attributed to the lower Lewis basicity of esters, precluding the formation of a catalytically active species, as observed by [1]H NMR spectroscopy (see Supporting Information). We therefore shifted our attention to the more reactive, conformationally locked cyclic esters, which readily underwent double hydroboration with 5 mol% catalyst loading and 2.1 equivalents of HBpin under mild conditions (Fig. 2b). A variety of ring sizes were investigated ($n$ = 5, 6 and 7) with γ-butyrolactone, δ-valerolactone and ε-caprolactone all undergoing smooth reduction to afford the corresponding borylated diols (**4a, 4b** and **4c**, respectively) in excellent yields. Reduction of phthalide and 3-isochromanone was also achievable, albeit at a higher temperature (50 °C) and longer reaction times.

Finally, we directed our focus to the hydroboration of carbon dioxide. Hydroboration proceeded in a J. Youngs NMR tube loaded with HBpin, 1 atm dry $CO_2$ and 2.5 mol% of **A** in $C_6D_6$ (0.6 ml); the reaction was monitored by [1]H and [11]B NMR spectroscopy, with partial consumption of HBpin at ambient temperature within 6 h being observed. Analysis of the [1]H NMR spectroscopic data revealed the formation of Bpin derivatives of formoxy (**5**), acetal (**6**) and methanol (**7**), Table 1. Using hexamethylbenzene as an internal standard to calculate solution yield, we observe a very slight selectivity towards **6** and **7**, both yielding 14%, closely followed by **5** with 10% conversation. In [11]B NMR spectroscopy, we also observe the presence of O(BR$_2$)$_2$. Due to the only modest conversion of HBpin and no clear selectivity to one product an alternative borane source, (9-BBN)$_2$ was screened under the same conditions described. Using (9-BBN)$_2$ with no catalyst led to no observable reduction products by [1]H NMR spectroscopy, even after extended reaction times (72 h); however, when the reaction was repeated with 2.5 mol% of **A**, full consumption of the borane is observed using [11]B NMR spectroscopy within 15 min. From the [1]H NMR spectroscopic data, adduct **7** was exclusively observed in 66% yield. Encouraged by these results, we explored the possibility of reducing the catalyst loading of **A** to 1 mol%. At ambient temperature, (9-BBN)$_2$ was consumed within 1 h, leading to product distribution comparable with the higher catalyst loading, yielding 65% of **7** with some O(BR$_2$)$_2$ observed by [11]B NMR spectroscopy. To confirm results, each experiment was repeated, giving comparable results as seen in the supporting information.

### Kinetic Studies

To further ascertain mechanistic insights into the hydroboration of ketones, a kinetic investigation was undertaken. Cyclohexylphenyl ketone was selected as the representative substrate due to its slower reaction rates. Initial reactions were monitored by [1]H NMR spectroscopy following the growth of the hydroboration product, **2i** at 273 K using the standard reaction conditions of 1 mol% complex **A** and a 1:1.1

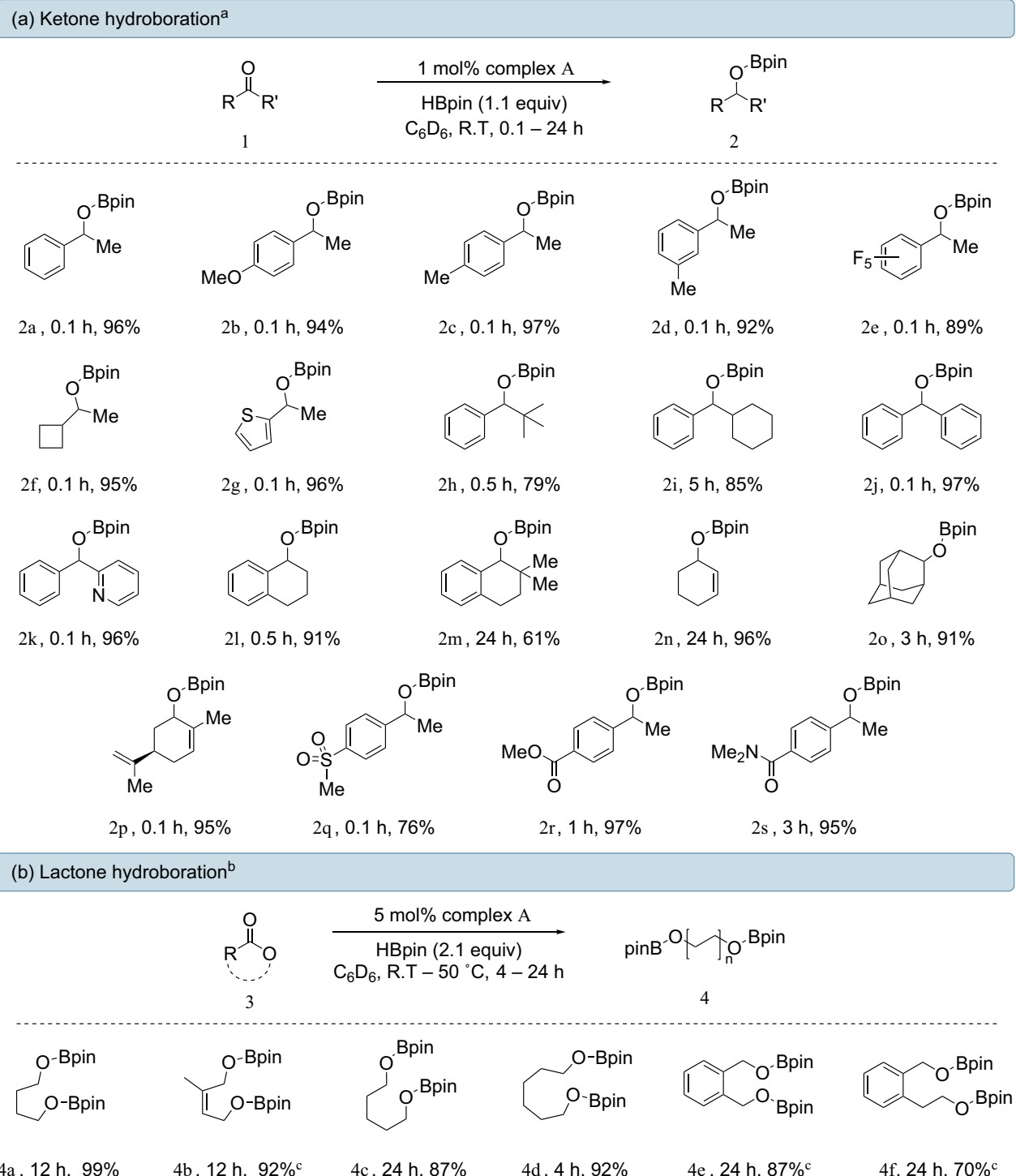

**Fig. 2 | Substrate scope for the hydroboration.** Hydroboration scope of **a** ketones and **b** lactones. All yields given are isolated yields. [a] Reactions were performed with **1a–1o** (0.205 mmol), HBpin (0.225 mmol, 1.1 equiv), Complex **A** (1.0 mol%), in 0.6 mL of benzene-$d_6$. [b] Reactions were performed with **3a–3f** (0.103 mmol), HBpin (0.225 mmol, 2.1 equiv), Complex **A** (5.0 mol%), in 0.6 mL of benzene-$d_6$. [c] 50 °C.

ratio of ketone (0.313 M) to HBpin (0.344 M), (Fig. 3). A fractional dependence on [**A**] is deduced through the variation of the catalyst loading whilst keeping the concentrations of ketone and HBpin constant (Fig. 3a). This dependency is suggestive of a dissociative process where one equivalent of an 'active' and one equivalent of an 'inactive' monomer is generated, as observed by the linear dependency of $K_{obs}$ vs [**A**] (see supporting information). This slightly higher than half-order

dependency can be attributed to an equilibrium process where the inactive monomer dissociates an $N_2$ moiety and can then engage in catalysis. Variation of the [ketone] revealed a major dependence on the reaction stoichiometry with maximum saturation in ketone concentration limits reminiscent of enzymatic kinetics (Fig. 3b). This plateauing of the reaction rate at high ketone concentrations could be attributed to the competing formation of an off-cycle ketone-iron

## Table 1 | Iron catalyzed hydroboration of $CO_2$

| Carbon dioxide hydroboration[a] | | | |
| --- | --- | --- | --- |

$CO_2$ → (x mol% complex **A**, 0.205 mmol $HBR_2$, $C_6D_6$, R.T, 1–7 h) → $R_2BO$CHO **5**, $R_2BO$OBR_2 **6**, $H_3C$OBR_2 **7**, $R_2B$O$BR_2$ **8**

| Borane | A (mol%) | 5 | 6 | 7 |
| --- | --- | --- | --- | --- |
| HBpin | 2.5 | 10 | 14 | 14 |
| (9-BBN)$_2$ | 0 | – | – | – |
| (9-BBN)$_2$ | 2.5 | – | – | 66 |
| (9-BBN)$_2$ | 1 | – | – | 65 |

[a] Reactions were performed with $CO_2$ (1 atm), HBpin (0.205 mmol), Complex **A** (x mol%), in 0.6 mL of benzene-$d_6$.

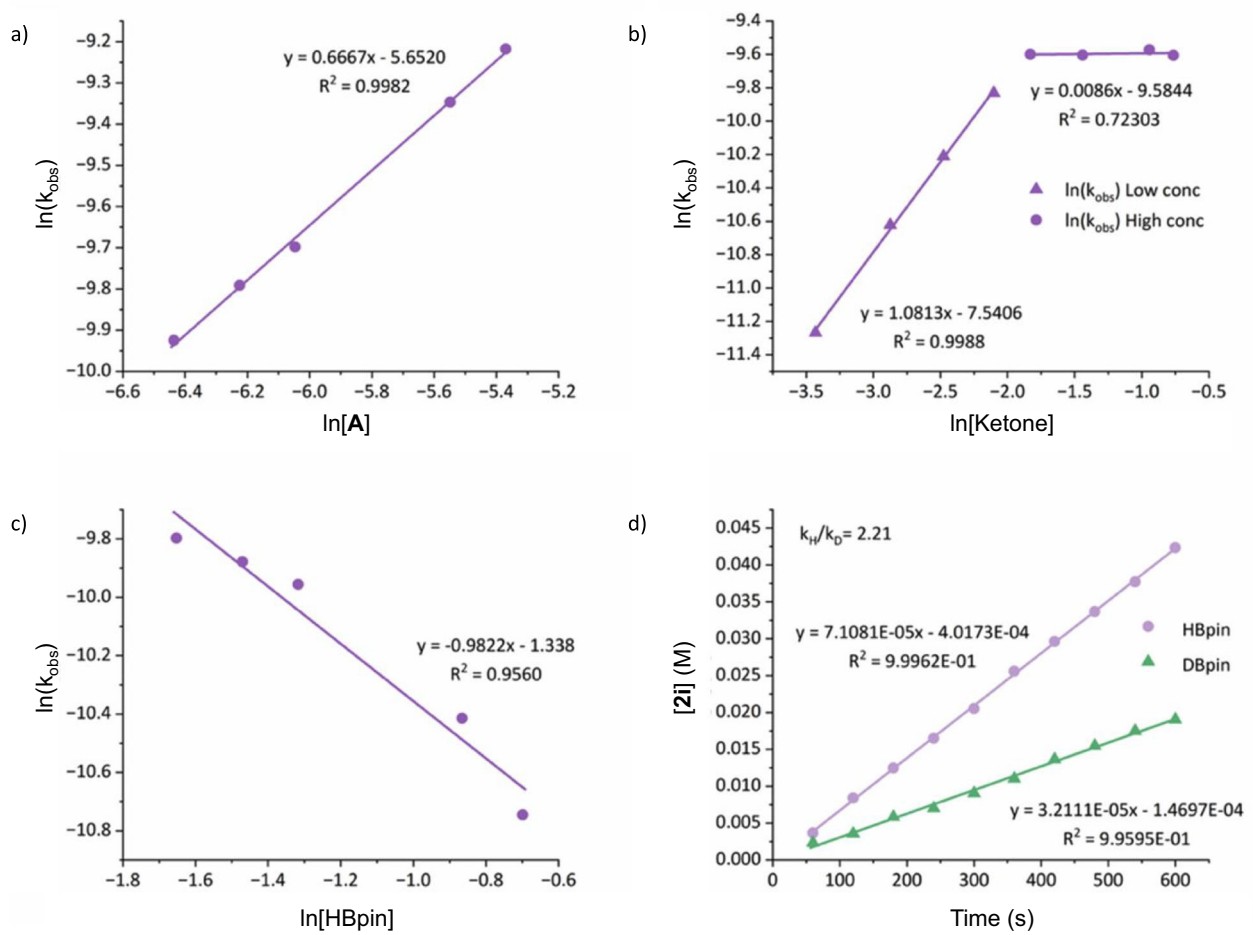

**Fig. 3 | Kinetic data for mechanistic determination. a** Rate vs concentration plot for the reaction rate law order in [complex **A**]; **b** rate vs concentration plot for the reaction rate law order in [ketone]; **c** rate vs concentration plot for the reaction rate law order in [HBpin]; **d** Initial rates of ketone hydroboration with HBpin and DBpin for KIE determination.

complex. Unfortunately, attempts to isolate or identify this putative species proved unsuccessful. At low concentrations of ketone, a *pseudo*-first-order dependency is observed, suggestive of its participation in the rate-determining step prior to potential catalyst saturation. Finally, the order in [HBpin] was determined by variation of the [HBpin] under standard reaction conditions and indicated an inverse

dependency on HBpin (Fig. 3c), implying kinetic inhibition is competing with the turn-over limiting step. To elucidate the nature of this inverse order in HBpin, we explored the reaction of HBpin with both the ketone and complex **A**. Notably, [1]H and [11]B NMR spectroscopy provided no evidence for a ketone-HBpin adduct nor an adduct of complex **A**·HBpin. These data indicate a more complex process is in

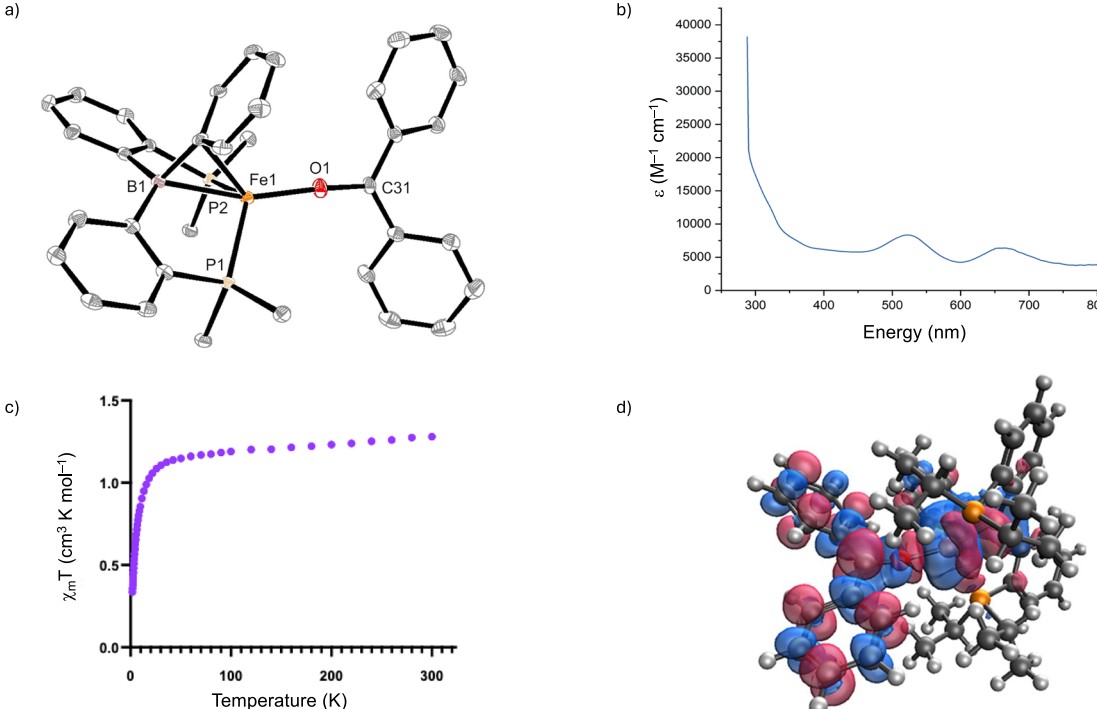

**Fig. 4 | Experimental and computational evaluation of complex 9. a** Solid-state structure of complex **9**. Ellipsoids are drawn at the 50% probability level; methyl groups and hydrogen atoms on isopropyl groups have been omitted for clarity. Selected distances of **9** (Å): Fe–P1 2.3760(8), Fe–P2 2.3424(8), Fe–O1 1.8364(18), Fe–B 2.273(3), O–C31 1.295(3); **b** UV-Vis spectrum of **9** in toluene; **c** Temperature-dependent SQUID magnetic susceptibility, $\chi_m T$ vs $T$; **d** distribution of positive (red) and negative (blue) spin density of complex **9** (isodensity = 0.02 Å$^{-3}$) from the $\omega$B97X-D3/def2-SVP DFT calculations (<S$^2$> = 3.28).

operation other than a two-component assembly and are suggestive of a three-component assembly between the iron, ketone and HBpin in the rate-determining step. Interestingly, a strong dependence on the order of substrate addition was observed. For the substrate scope and kinetic experiments, the ketone and **A** were pre-mixed, followed by the addition of HBpin. We observed that when the ketone and HBpin are pre-mixed prior to the addition of **A**, there is a considerable decrease in the rate from $7.1 \times 10^{-5}$ to $2.72 \times 10^{-5}$ M/s, respectively. This reinforces the inverse order dependency observed for HBpin, implying a competitive inhibition reaction preventing the formation of the active catalyst.

A comparison of the pseudo-first order rate constants for hydroboration of cyclohexylphenyl ketone in which HBpin was exchanged for DBpin provided a kinetic isotope effect (KIE, $k_H/k_D$) of 2.21 (Fig. 3d). Although considered a relatively small KIE value, this constitutes a large secondary kinetic isotope effect and is comparable to results ($k_H/k_D = 1.62$) reported by Hill et al., during studies of magnesium-catalyzed hydroboration, in which B–H bond cleavage is part of the rate-determining step[52].

**Stoichiometric Studies**

To gain insight into these catalytic reactions, a series of stoichiometric reactions were conducted. Monitoring of NMR-scale reactions performed using cyclohexylphenyl ketone was particularly informative. We previously reported that **A** reacts with an excess of HBpin to generate an iron(II) dihydride species (**B**), along with B$_2$pin$_2$ and PhBpin within 6 h at 50 °C, with no reaction observed at room temperature[53]. When **B** was reacted with an equimolar quantity of cyclohexylphenyl ketone, no reaction was observed at room temperature after 16 h. These results provoked a more in-depth investigation into how **A** interacts with substrates. We were particularly interested in how the carbonyl motif of the ketone may coordinate to the iron center (i.e., $\eta^1$-

through the oxygen or $\eta^2$ through the $\pi$-bond, as reported for the analogous Ni-complex)[53].

A reaction using an equimolar quantity of cyclohexylphenyl ketone resulted in partial consumption of **A** to generate a new paramagnetic species, evidenced by $^1$H NMR spectroscopy; however, only a small amount was converted and was not stable to isolation. Even in catalytic equivalents, complete conversion of complex **A** into a new species is not observed, which could be attributed to the half-order dependency observed in kinetic analysis. When ketones bearing aromatic substituents were mixed with complex **A**, a color change from red to purple occurred and new paramagnetic species were observed by $^1$H NMR spectroscopy; however, these quickly decomposed upon isolation. Addition of benzophenone to complex **A** in C$_6$D$_6$ also resulted in an immediate color change from dark red to intense purple. $^1$H NMR spectroscopy showed full conversion of **A** with 15 new paramagnetically shifted resonances, indicating $C_s$ symmetry in solution. An iron-ketyl radical complex, [($^{iPr}$DPB$^{Ph}$)Fe(OC(Ph)$_2$)] (**9**), was isolated, and crystals suitable for X-ray crystallographic studies were grown from slow evaporation of diethyl ether (Fig. 4a). The benzophenone is coordinated to the iron center exclusively via the oxygen, resulting in a pseudo-tetrahedral iron center ($\tau_4$ = 0.71). Upon analysis of bond metrics, an elongation of the C–O bond length (1.295(3) Å) in comparison to the free benzophenone C = O bond length (1.2233(17) Å) was observed. This bond length is comparable to that observed for an authenticated neutral iron(II)-benzophenone ketyl radical (1.2989(19) Å) reported by Gade[54]. Equally, the Fe–O bond length of 1.8364(18) Å is similar to the only other reported iron-ketyl complexes by Gade[54] and Werncke[55] (Fe(II)–O 1.8565(10) Å and 1.869(1) Å) and is in good accordance with an Fe(I) species bearing an anionic oxygen donor ligand[56].

UV-Vis spectroscopic analysis of **9** showed two distinct absorptions at 522 nm and 652 nm, which is consistent with previously

a)

Yield of 2j with styrene (100 mol%), 96%
Yield of 2j with BHT (100 mol%), 95%

b)

2t 94%    0%

**Fig. 5 | Reactions to probe if the hydroboration proceeds through a single electron or two-electron pathway. a** Radical inhibition experiments; **b** radical clock experiments. All reported yields are isolated yields.

reported benzophenone ketyl radicals (Fig. 4b)[55,56]. From TD-DFT calculations, the absorption at 522 nm is attributed to the π(ketyl) to π*(borane) transitions (computed as 490 nm), whereas the absorption at 652 nm (TDDFT value of 605 nm) is attributed to the π(ketyl) to π*(borane) and π(ketyl) to π*(ketyl) transitions, which both share considerable ligand to metal charge-transfer (LMCT) character (see supporting information). X-band EPR-spectroscopy was performed at 5 K in toluene glass to try and observe a $g$-value around 2.00, indicative of the presence of the organic ketyl radical. No such absorptions were observed, suggesting complex **9** has an overall whole integer spin.

To rationalize the absence of the $g = 2.00$ EPR absorption and gain further insight into the electronic structure of **9**, their magnetic properties in both solution (Evans method) and the solid state were probed. The effective magnetic moment ($\mu_{eff} = 2.76\ \mu_B$) in solution is in good agreement with the value obtained in the solid state at 300 K ($\mu_{eff} = 3.12\ \mu_B$). The magnetic data are consistent with what would be expected for an S = 1 system. The $\chi_m T$ vs. $T$ slope depicts a steady decrease to 50 K followed by a sharp drop from 50 K to 2 K (Fig. 4c). Interestingly, the $\chi_m T$ vs $T$ curve does not plateau at higher temperatures suggesting the presence of antiferromagnetic coupling between an unpaired electron on the iron and the ketyl radical. To understand how we obtain an S = 1 system with antiferromagnetic coupling, DFT calculations were performed using both the ωB97X-D3 and CAM-B3LYP levels of theory, along with the def2-SVP basis set. SCF stability calculations consistently show a broken spin-symmetry solution, corresponding to four unpaired electrons, with antiferromagnetic coupling. The resulting spin density is shown in Fig. 4d. A Natural Bond Orbital (NBO) analysis reveals three unpaired electrons on the Fe center delocalized across the $d_{xy}$, $d_{xz}$, $d_{x^2-y^2}$ and $d_{z^2}$ orbitals, with antiferromagnetic coupling to the unpaired electron on the ketyl radical moiety. Calculated values of the approximate antiferromagnetic coupling gave a value of $-2071\ cm^{-1}$, qualitatively indicating strong antiferromagnetic coupling between the ketyl and Fe sites.

To ascertain if the iron(I)-ketyl radicals are part of the on-cycle pathway, complex **9** was subjected to 10 equivalents of HBpin under standard reaction conditions and monitored by [1]H NMR spectroscopy. After 25 min, only a trace amount of hydroborated product **2j** was observed. After 16 h, complete conversion of complex **9** to **2j** and complex **A** was observed, suggesting the ketyl-radical is in equilibrium with the carbonyl-bound species. To further probe the possibility of a radical pathway, inhibition experiments were conducted using the radical scavenger styrene and the radical inhibitor BHT under standard conditions. No decrease in the yield of **2j** was observed in both cases, suggesting that a radical pathway is not in operation. To further probe the involvement of radicals in this transformation, a radical clock experiment was conducted. Radical ring-opening products were not observed, whilst the hydroborated product **2p** is isolated in 94% yield, further suggesting a two-electron pathway is in operation (Fig. 5).

## DFT studies

Dimeric pre-catalyst complex **A** initially coordinates HBpin, before the dimer splits into **11** (with HBpin) and **12** (with N$_2$ bound). The ketone (**2i**) then forms a B–O bond with the bound Bpin, before a Fe-mediated ligand to ligand hydride transfer (LLHT), before elimination of the hydroborated product to reform **A**, which re-enters the catalytic cycle. Following the formation of the B–O bond, the rate-limiting step is the hydride transfer (Fig. 6, ‡11–13 (2)), supported by the KIE data presented earlier. The barrier is ~5 kcal mol$^{-1}$ lower in energy than the equivalent transition state for HBpin and the ketone reacting together in the absence of **A**. At the transition state of **11–13**, the out-of-plane B–H bond is responsible for the higher reactivity, forming the new B–C bond, while coordination to the Fe complex reduces the energy barrier for the hydride transfer. Coordination of **12** to this complex appears to be essential in reducing this barrier and supports the hypothesis from the kinetic data that only a fractional quantity of the Fe centers in **A** are involved in the catalytic step; the transition state is >5 kcal mol$^{-1}$ higher without **12** present. The energies of the different minima with different spin multiplicities are given in Table S3.

Alternative pathways were considered, where in all cases the hydride transfer step was significantly higher than the LLHT barrier height (Fig. S10 in the Supporting Information). In the alternative to the LLHT step, the (rate-limiting) barrier height was calculated to be 29.1 kcal mol$^{-1}$, while in the pathway in which the ketone coordinates to the Fe complex first, the rate-limiting barrier height is 31.6 kcal mol$^{-1}$ (top in the alternative pathway, Fig. S10) and 38.2 kcal mol$^{-1}$ (bottom in the alternative pathway, Fig. S10). Based on the kinetic data obtained, DFT results, and previous mechanistic studies conducted using iron-diphosphinoborane complexes, a plausible catalytic cycle can be tentatively assigned for cyclohexylphenyl ketone (Fig. 7)[51,53,56,57].

In summary, complex **A** has been demonstrated as an active pre-catalyst for hydroboration of a broad range of ketones, cyclic esters, and CO$_2$ to form boryl ethers under mild conditions. The hydrofunctionalization of CO$_2$ by (9-BBN)$_2$ led to the selective formation of methoxy borane in less than 1 h at ambient temperature. Stoichiometric reactions have shown that in the case of sterically unhindered ketones, the substrate coordinates to the iron center with a loss of N$_2$, and this is believed to be the entry point in the catalysis. Kinetic studies with cyclohexylphenyl ketone demonstrated an inverse dependency for HBpin and a KIE value of 2.21, indicating that B–H cleavage is likely the rate-determining step. From DFT studies, the hydride transfer step proceeds via direct LLHT, which is mediated by the iron. An unreported iron(I)-ketyl radical was isolated from the reaction of **A** with benzophenone, with magnetic measurements and DFT studies revealing an S = 1 iron center containing three unpaired electrons on iron and an antiferromagnetically coupled ketyl radical.

## Methods
### Provided here are key protocols
Complete experimental details (general considerations, synthesis & characterization data, catalytic reactions, reactivity studies, kinetics

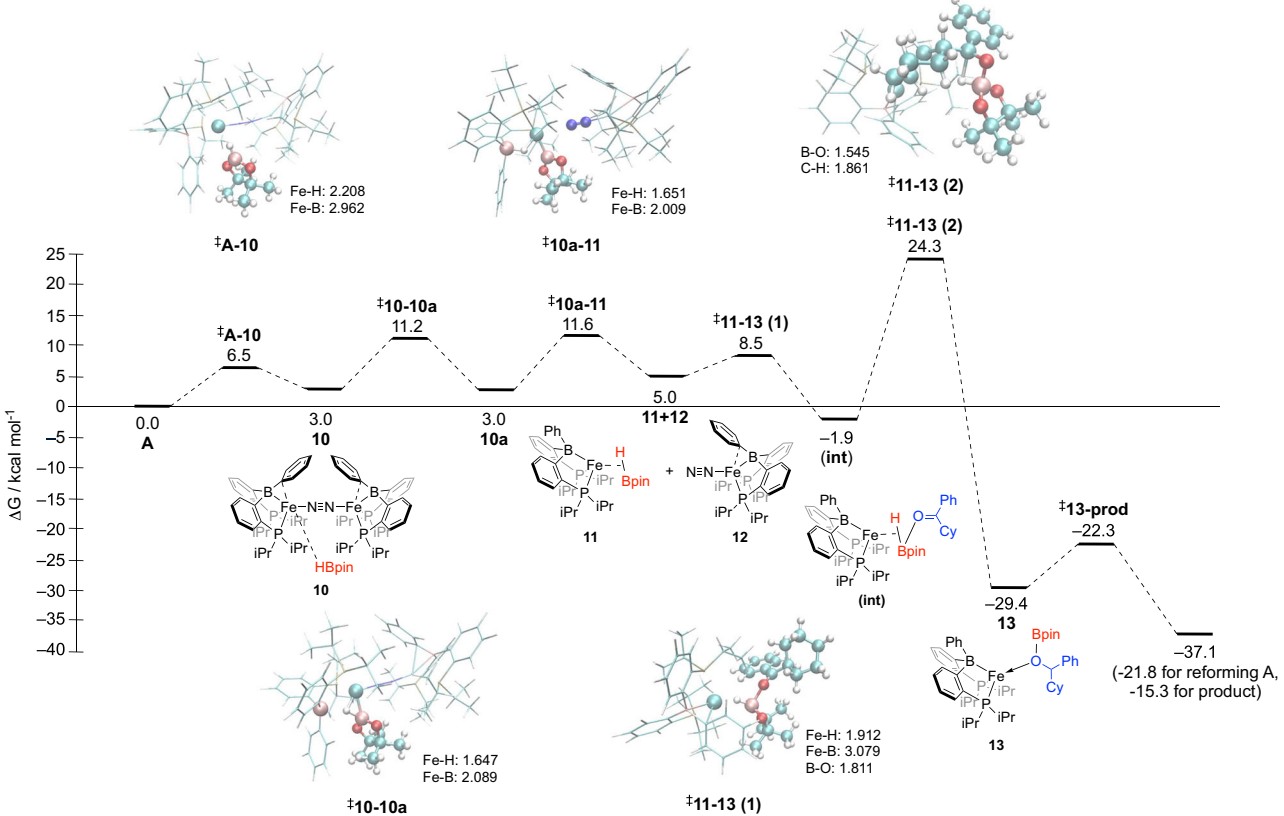

**Fig. 6 | Gibbs free energy profile.** Calculated Gibbs free energy profile for the hydroboration of cyclohexylphenylketone by HBpin in the triplet state. Energies are from ωB97X-D3/def2-TZVP//r2SCAN−3c calculations, with r2SCAN−3c enthalpy and translational entropy corrections.

measurements) and further computational details are provided in the Supplementary Information.

## Synthesis of [{(iPrDPBPh)Fe}2(μ-1,2-N2)] (Complex A)

The iPrDPBPh ligand (1.33 g, 2.81 mmol) and FeBr2 (0.602 g, 2.81 mmol) were stirred at R.T. in THF (80 mL) until all solids dissolved into a yellow solution. Volatiles were removed in vacuo, generating yellow/orange solids, this was left under a vacuum for 30 min after all solvents were visibly removed. Et2O (100 mL) was added to the solids and stirred vigorously for 1 h to produce a bright yellow precipitate, volatiles were removed in vacuo and the remaining solids were subsequently dissolved in benzene (100 mL) and allowed to stir overnight before being added to a 1% Na/Hg amalgam (Na: 291 mg, 0.013 mol) and left to stir for 16 h at R.T. The dark red solution was filtered through Celite® and volatiles were removed in vacuo. Dark brown/red solids were washed in pentane (20 mL), solids were dried *in vacuo* before being washed in cold Et2O (2 × 10 mL) (0.974 g, 1.83 mmol, 65%). ¹H NMR (400 MHz, C6D6) δ 135.00, 44.52, 34.77, 28.38, 26.50, 7.46, 0.32, −1.45, −2.28, −6.25, −9.23, −76.98 ppm.

## Synthesis of [(iPrDPBPh)Fe(OC(Ph)2)] (9)

Complex 9 was prepared by dissolving complex A (82 mg, 0.150 mmol) and benzophenone (27.2 mg, 0.150 mmol) in C6D6 (0.6 ml). Immediately, the color changed from dark red to dark purple at R.T., volatiles were removed in vacuo, affording 9 as a dark solid. Slow evaporation from a concentrated solution of diethyl ether gave dark purple single crystals suitable for XRD (101 mg, 0.141 mmol, 94%). ¹H NMR (400 MHz, C6D6) δ 132.20, 126.21, 48.60, 37.76, 35.20, 22.83, 4.96, 2.41, 0.02, −3.78, −5.77, −8.79, −14.25, −29.77, −31.28. ATR-IR (cm⁻¹) 1359 (C–O). Solution magnetic moment (25 °C, C6D6) 2.76 μB. UV-vis [toluene, λ(nm) {ε(M-1 cm⁻¹)}]: 522(8345), 622(6376).

## General procedure 1: hydroboration of ketones

In a nitrogen-filled glovebox, an oven-dried J-Youngs NMR tube was charged with [{(iPrDPBPh)Fe}2(μ-1,2-N2)] (0.00206 mmol), C6D6 (0.6 mL), substrate (0.205 mmol), HBpin (0.225 mmol), followed by toluene (0.205 mmol) as an internal standard for NMR quantification. The reaction mixture was left at R.T. until reaction was complete (0.1–24 h), monitored by ¹H NMR spectroscopy. Volatiles were removed *in vacuo*, the mixture was suspended in Et2O and filtered through a short plug of Celite® in a glove box, and volatiles were removed *in vacuo* to reveal the isolated product.

## General procedure 2: hydroboration of cyclic esters

In a nitrogen-filled glovebox, an oven-dried J-Youngs NMR tube was charged with [{(iPrDPBPh)Fe}2(μ-1,2-N2)] (0.005 mmol), C6D6 (0.6 mL), substrate (0.103 mmol), HBpin (0.225 mmol), followed by toluene (0.103 mmol) as an internal standard for NMR quantification. The reaction mixture was either left at R.T. or added to an oil bath (50 °C) for 4–48 h and monitored by ¹H NMR spectroscopy. Volatiles were removed *in vacuo*, the mixture was suspended in Et2O and filtered through a short plug of Celite® in a glove box, and volatiles were removed *in vacuo* to reveal the isolated product.

## General procedure 3: hydroboration of carbon dioxide

In a nitrogen-filled glovebox, the desired amount of [{(iPrDPBPh)Fe}2(μ-1,2-N2)] 2.5 mol%–1 mol% was dissolved in 0.6 mL of C6D6 with HBR2 (0.205 mmol) and hexamethylbenzene as an internal standard for NMR quantification in a J-Youngs NMR tube. The NMR tube was taken out of the glovebox and degassed by freeze-pump-thaw, then backfilled with CO2 (1 bar) from a cylinder. Yields were calculated by ¹H NMR spectroscopy.

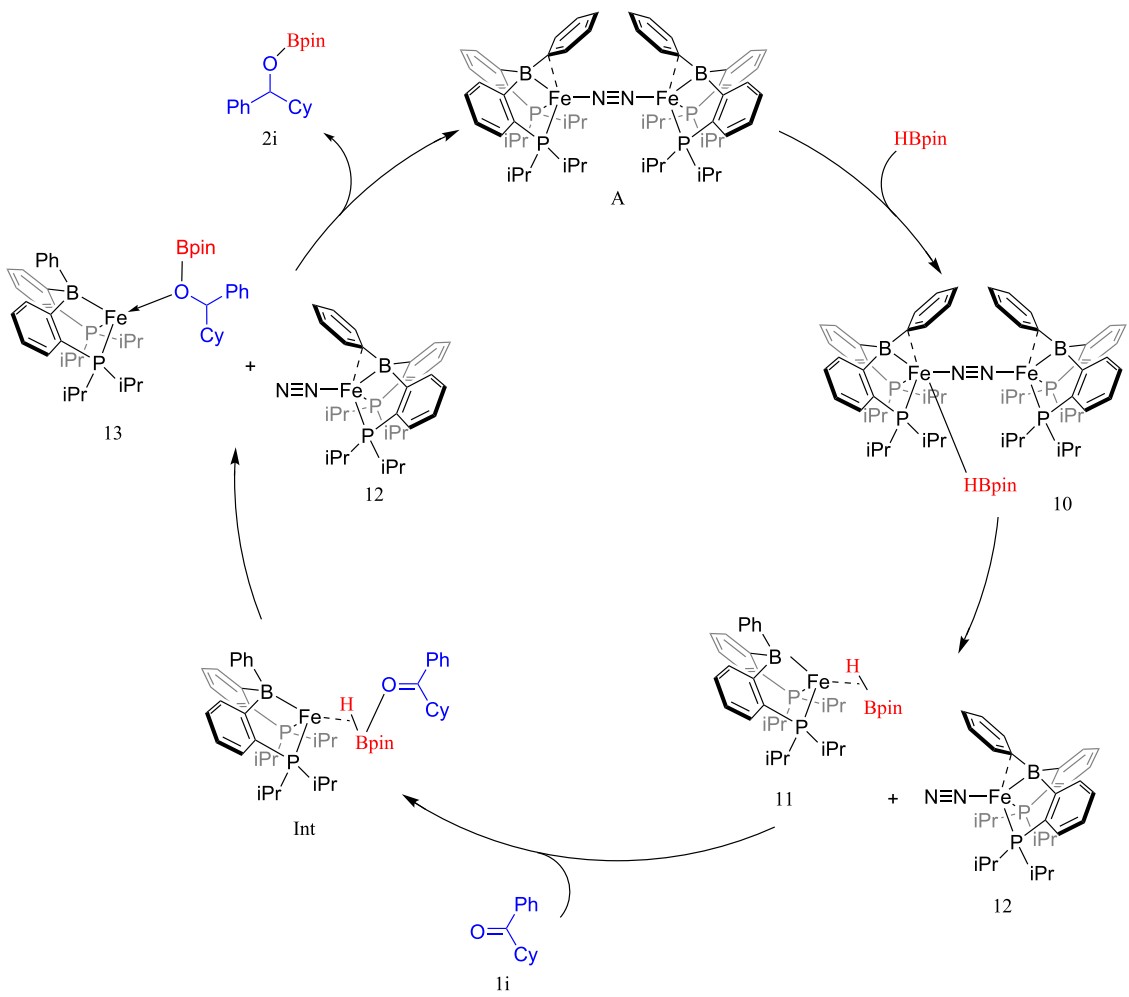

**Fig. 7 | Proposed catalytic cycle for the current work.** A proposed catalytic cycle for this work based on the kinetic and computational data obtained.

## Data availability

All data generated in this study are provided in the Supplementary Information and Source Data file. The X-ray crystallographic data for structures reported in this study have been deposited at the Cambridge Crystallographic Data Center, under deposition numbers 2339051 (**9**). Copies of the data can be obtained free of charge via https://www.ccdc.cam.ac.uk/structures/. All data are available from the corresponding author upon request. Source data are provided with this paper.

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

## Acknowledgments

We gratefully acknowledge the UK EPSRC (DW: EP/W023172/1). We thank the University of Manchester (EPSRC DTG) for a studentship (LAG), the EPSRC UK National Electron Paramagnetic Resonance Facility for access to SQUID magnetometry and EPR (EP/W014521/1, EP/W014521/1). We also thank Dr Ralph Adams and Dr Aminata Sakho for NMR training, Dr Bono Van Ijzendoorn for free use of the CO$_2$ cylinder. We also thank Dr. John Seed for help with sample sealing and Prof. Eric McInnes for help interpreting the magnetic data. David Robinson thanks NTU for the provision of the Avicenna high-performance computing cluster on which the DFT calculations were performed.

## Author contributions

L.A.G., R.J.S., and D.W. formulated the project. L.A.G. and R.J.S. synthesized the materials and measured and analysed all NMR, IR, and UV-Vis spectroscopy and single crystal X-ray diffraction data. L.A.G. performed and analysed all kinetic data. L.A.G. prepared samples for SQUID and EPR measurements, A.B. performed the SQUID and EPR experiments, with the data analyzed by A.B. and D.W. D.R. performed all DFT calculations. L.A.G., D.R., and D.W. wrote the manuscript, and all authors contributed to revising the manuscript.

## Competing interests

The authors declare no competing interests.
