## [Transparent Peer Review file · Nature Communications]

Iron-Borane Catalyzed Carbonyl Hydroboration and Isolation of an Iron(I)-Ketyl Radical

Corresponding Author: Dr Darren Willcox

Version 0:

Reviewer comments:

Reviewer #1

(Remarks to the Author)

The work by Willcox and Robinson describes the use of an iron(0) metalborane complex as a precatalyst for hydroboration of ketones, cyclic esters and CO₂ under mild conditions. Mechanistic investigations, including DFT calculations, suggest that ketone hydroboration likely proceeds via an Fe-mediated ligand-to-ligand hydride transfer (LLHT) mechanism. Notably, the authors have isolated an off-cycle iron(I)-benzophenone ketyl radical complex with an S = 1 antiferromagnetic ground state, offering valuable mechanistic insights. Although the hydroboration of carbonyls has been extensively studied, the authors claim an "unprecedented Fe-mediated LLHT mechanism"; however, LLHT pathways are actually quite common in metal-catalyzed hydroboration reactions.

My primary concern is a mechanistic inconsistency: the authors report that complex A does not react directly with HBpin at room temperature but does activate benzophenone to generate the ketyl radical. Nevertheless, the mechanism proposed in Figure 2 depends on HBpin activation, which appears contradictory.

In my opinion, the most novel aspect of this work is the direct activation of benzophenone to form the iron(I)-benzophenone ketyl radical, with significant implications for catalytic transformations. While this study is interesting and potentially suitable for publication in Nature Communications, further experimental and computational studies are needed to clarify the initiation step—specifically, whether catalysis is triggered by substrate activation or HBpin activation.

More scientific considerations:

1. The claim of an "unprecedented Fe-mediated LLHT mechanism" should be tempered, as Fe–H transfer to a carbonyl carbon is a well-known process in transition metal-catalyzed hydroboration.
2. The Figure 1 caption refers to a "van't Hoff plot," but the data illustrate k_{obs} versus substrate/catalyst concentration. Please correct this mislabeling.
3. For CO₂ hydroboration, control experiments with (9-BBN)₂ should be provided to establish baseline reactivity.
4. The stated "1:1 ratio of ketone (0.313 M) to HBpin (0.344 M)" is inaccurate and should be clarified or corrected.
5. The explanation, "At low ketone concentrations a pseudo first-order dependency is observed suggesting that the ketone is required up to and including the rate-determining step," is confusing and should be revised for clarity.
6. Please address whether complex A (Fe(0)) could undergo oxidative addition with HBpin to form Fe–H/Fe–Bpin, with Fe–H as the active catalyst. Alternatively, is cooperative H–B activation by Fe–B in intermediate 11 via a four-membered transition state feasible for hydride/Bpin transfer to the carbonyl?
7. In Scheme 4, correct the labeling of intermediates 11 and 12 from "PH". Also, ensure that panels a, b, c, d in Figure 1 are properly labeled, and check the designation for compound 2i.
8. The mechanistic section is difficult to follow; additional calculations and clarification are needed regarding the initiation

step.

9. For the monomeric Fe(I) species in Figure 3, did the DFT calculations consider all relevant spin states (doublet, quartet, sextet)? Please report the relative energies in the Supporting Information.

10. What is the activation barrier for the background (uncatalyzed) process?

11. In the proposed rate-limiting transition state, the metal appears to have weak interactions with the substrates. Is this plausible? The computed barrier of 26.2 kcal/mol suggests the reaction may not occur at room temperature.

12. All optimized coordinates should be included in the Supporting Information.

Reviewer #2

(Remarks to the Author)

The manuscript by Willcox, Robinson and coworkers reports on an iron metalloborane dinitrogen complex that catalyzed the hydroboration of ketones, cyclic esters and carbon dioxide. This is a nice piece of work that represents a detailed study on earth-abundant iron-catalyzed reactions with a range of carbonyl substrates. Both experimental and computational work has been conducted for uncovering the mechanistic aspect, along with the isolation and characterization of a novel iron-ketyl radical complex resulting from the stoichiometric reactions. While this reviewer is mostly positive on the overall quality of the work, the following factors diminish my enthusiasm about its publication in this high impact interdisciplinary journal. A more specialized journal in catalysis would be better suited.

1, both the catalyst and the catalytic reactions presented here are not new and in fact the hydroboration of carbonyl compounds has been long explored with numerous catalysts or even without a catalyst in the literature. The mechanistic study is much appreciated, yet it does not represent any conceptual advances that justify a publication in Nature Communications. The isolation of an iron-ketyl radical product was significant, but turned out not to be a key intermediate in the catalytic cycle. Thus, it did not benefit the mechanistic understanding.

2, as for the catalytic reactions, the substrate scope was not well studied. More challenging substrates containing reducible functional groups should be included. It was also unfortunate to display poor catalytic activity towards general ester substrates.

Reviewer #3

(Remarks to the Author)

The article by Grose et al. describes an experimental and computational study into very interesting iron boron chemistry for hydroboration of carbonyl containing compounds, and they describe the isolation of a unique iron(I) ketyl complex in an antiferromagnetic ground state. The manuscript is well written and the topic is of high scientific interest.

Overall the work is highly interesting both in terms of experiment and theory.

I will focus my review on the computational work in this study and I ask the authors to consider the comments below:

1) In iron coordination compounds, the electronic structure may have a large flexibility in spin states depending on the detailed surroundings. This is apparent in the intricate electronic structure found for antiferromagnetically coupled complex 9, but it will also play a role in the dinuclear N₂-bridged complex A. Assuming S=1 for the two Fe centres, it is not obvious how the magnetic moments of the two iron centres are coupled and what the electronic ground-state structure looks like. Further, in terms of reaction kinetics and thermodynamics the concept of two-state or multi-state reactivity involving spin crossover steps is often relevant. Therefore the electronic structure of all Fe containing species must be carefully evaluated in order to ascertain the ground-state potential energy surface.

2) With regard to reactivity the entropic contributions cannot be neglected in my opinion. This is particularly true for reaction steps involving the cleavage/formation of dimeric structures or, in general, steps involving change in particle numbers.

3) I would be interested to learn more about the authors' perspective on why the iron ketyl radical 9 exhibits such unusual antiferromagnetic features. Are there particular properties of the iron-boron interaction and/or the iron-carbonyl bond that govern the antiferromagnetic ground state? Or conversely, what inhibits this antiferromagnetic behaviour in other iron-ketyl compounds? Further elaboration would enhance the impact of the manuscript.

Further specific remarks

Main text:

Computational study of 9:

How does the computed molecular structure/geometry compare with that extracted from XRD?

Please report the relative energies of the neighbouring (pure) spin multiplicities, S=2 quintet and S=0 closed-shell singlet.

Please state the isodensity value for the spin density plot in Figure 2d.

Please state the $\langle S^2 \rangle$ value for 9 also somewhere in the main text (e.g. in the caption of Figure 2d).

The spin populations (NPA or Mulliken) on the relevant atomic sites (Fe, B, Fe-coordinated phenyl, O, rest of the ketyl group) should be reported - in the ESI as appropriate.

Energy profile for hydroboration:

It is not clear which energy values are reported for the computed potential energy surface:

On the y-axis of Figure 3 " $\Delta H + ZPE$ " is given. This is a conflicting description. ΔH would already include ZPE. Perhaps " $\Delta E + ZPE$ " is meant? Then these values would correspond to zero K data. Is this the case here?

Please report Gibbs energies including entropy contributions at r.t. instead of just enthalpies (or energies at zero K). This is particularly important for comparison with experiment and for assessment whether a barrier height is rate limiting or not.

If I understand correctly, only the $S=1$ triplet spin surface is considered. Other spin multiplicities might play a role here, and these should be investigated and discussed in the same detail (or excluded if applicable).

Supporting information:

Please report the corresponding transition wavelengths/energies of the TD-DFT results. How does the computed spectrum compare to the experimentally observed one?

Figures S8-S10:

It is not clear which spin surface the PES refers to. Please report specifics on the electronic structure, at least the spin multiplicity of the species in the PES.

For comparison with experiment, Gibbs free energies at r.t. should be reported instead of "enthalpies plus ZPE".

The electronic structure of species A should be discussed in a little detail:

- How are the two magnetic iron centres coupled?
- Why are other spin multiplicities not considered or discussed?
- How is the possibility of spin change across the PES excluded?

Please carefully check and amend the References in the ESI.

Here are two examples:

Reference 13 seems incomplete.

The authors' list in Reference 15 for the QChem package doesn't look right.

Please include Cartesian coordinates, total energies and spin expectation values for all computed species.

Version 1:

Reviewer comments:

Reviewer #1

(Remarks to the Author)

The authors attempted to revise the manuscript in accordance with the reviewers' comments; however, the revisions and responses remain unsatisfactory.

1. The monomeric Fe(I) species can exist as a doublet, quartet, or sextet. However, the authors calculated only the quintet and triplet states. What is the total charge of this complex? Did the authors also calculate the Fe(II) species? If not, this omission raises significant concerns about the analysis.
2. According to the energy span model, the total barrier is calculated to be 26.2 kcal/mol, as noted in well-established literature within the field of computational catalysis (Accounts of Chemical Research, 2011, 44, 2, 101-110). There should be a more favorable reaction pathway that can satisfactorily account for the experimental observations. If the reaction occurs at room temperature, the barrier should ideally be below 25 kcal/mol.
3. The proposed mechanism suggests a concerted transition state involving a four-membered ring. Notably, there is no indication of coordination or bonding interaction between the metal center and the substrate in this transition state. What is the actual role of the catalyst in this scenario? This appears to be an unprecedented form of catalysis. The authors may well achieve a similar barrier without the catalyst.

Reviewer #2

(Remarks to the Author)

The revised manuscript has been greatly improved and the authors' responses to reviewers further convinced this reviewer to recommend this manuscript to be accepted for publication.

Reviewer #3

(Remarks to the Author)

All questions and concerns have been addressed, and I believe that the article is sufficient for publication.

Version 2:

Reviewer comments:

Reviewer #1

(Remarks to the Author)

The authors have properly addressed my questions and concerns, and the article can be accepted for publication.

REVIEWER COMMENTS

We thank all the reviewers for their comments and our responses to the rebuttals below are in purple. The appropriate changes have been made to the manuscript and SI and are identifiable as track changes throughout the documents.

Reviewer #1 (Remarks to the Author):

The work by Willcox and Robinson describes the use of an iron(0) metalborane complex as a precatalyst for hydroboration of ketones, cyclic esters and CO₂ under mild conditions. Mechanistic investigations, including DFT calculations, suggest that ketone hydroboration likely proceeds via an Fe-mediated ligand-to-ligand hydride transfer (LLHT) mechanism. Notably, the authors have isolated an off-cycle iron(I)-benzophenone ketyl radical complex with an S = 1 antiferromagnetic ground state, offering valuable mechanistic insights. Although the hydroboration of carbonyls has been extensively studied, the authors claim an "unprecedented Fe-mediated LLHT mechanism"; however, LLHT pathways are actually quite common in metal-catalyzed hydroboration reactions.

During the preparation of this manuscript, a detailed literature search revealed that there were no direct LLHT processes catalyzed by transition-metal complexes for carbonyl hydroboration, however, LLHT has been reported for s-block catalysts (*ACS Omega*, **2019**, *4*, 15893–15903). LLHT has been reported for Fe-catalyzed hydrosilylation of alkynes, however, this is reported to proceed via the formation of a discrete Fe(II) Fe-H/Fe-Si species (*J. Am. Chem. Soc.* **2020**, *142*, 16894–16902).

My primary concern is a mechanistic inconsistency: the authors report that complex A does not react directly with HBpin at room temperature but does activate benzophenone to generate the ketyl radical. Nevertheless, the mechanism proposed in Figure 2 depends on HBpin activation, which appears contradictory.

We thank the reviewer for pointing this out. We agree that the way we had drawn the proposed mechanism and the structures in the DFT calculated pathway are ambiguous, causing much of the confusion noted by reviewers below. During the DFT studies, the direct activation of HBpin by complex A was much higher in energy compared to the formation of the sigma-complex. We represented this sigma-complex as a 3-membered ring which could look like activation of the H–B bond. In the revised figure and scheme, this sigma-complex has been represented by a bond from the iron to the centre of the B–H bond to overcome this potential issue.

In my opinion, the most novel aspect of this work is the direct activation of benzophenone to form the iron(I)-benzophenone ketyl radical, with significant implications for catalytic transformations. While this study is interesting and potentially suitable for publication in *Nature Communications*, further experimental and computational studies are needed to clarify the initiation step—specifically, whether catalysis is triggered by substrate activation or HBpin activation.

From an experimental stand, no activation of the B–H bond is observed when HBpin is reacted with complex A at room temperature. Regarding the substrate activation, we see no formation of an Fe-Ketone species via ¹H NMR spectroscopy, with only complex A and the ketone resonances observable. This is also supported by a Job's plot type analysis where a flat line was observed meaning there is no complex formation between Fe and the ketone – presumably due to the steric bulk of the ketone, which is represented by the prolonged reaction time for this substrate. From the DFT calculations and the Job plot type analysis, a three-component transition state is required for the catalysis to occur, rather than specific activation of either the borane or the ketone.

More scientific considerations:

1. The claim of an "unprecedented Fe-mediated LLHT mechanism" should be tempered, as Fe–H transfer to a carbonyl carbon is a well-known process in transition metal-catalyzed hydroboration.

Whilst the formation of an Fe–H might be well-precedented in hydroboration chemistry (and we have previously reported this activation pathway for olefin and nitrile reduction), this reduction process transfers the H from HBpin directly to the ketone without the formation of a discrete Fe–H complex.

2.The Figure 1 caption refers to a "van't Hoff plot," but the data illustrate k_{obs} versus substrate/catalyst concentration. Please correct this mislabeling.

This has been changed in the manuscript.

3.For CO₂ hydroboration, control experiments with (9-BBN)₂ should be provided to establish baseline reactivity.

This has been added to the manuscript (Table 1 entry 2, Page 4)

4.The stated "1:1 ratio of ketone (0.313 M) to HBpin (0.344 M)" is inaccurate and should be clarified or corrected.

This has been corrected to 1:1.1

5.The explanation, "At low ketone concentrations a pseudo first-order dependency is observed suggesting that the ketone is required up to and including the rate-determining step," is confusing and should be revised for clarity.

We have revised this statement in the manuscript and replaced it with "At low concentrations of ketone, a pseudo first-order dependency is observed suggestive of its participation in the rate-determining step prior to potential catalyst saturation."

6.Please address whether complex A (Fe(0)) could undergo oxidative addition with HBpin to form Fe–H/Fe–Bpin, with Fe–H as the active catalyst. Alternatively, is cooperative H–B activation by Fe–B in intermediate 11 via a four-membered transition state feasible for hydride/Bpin transfer to the carbonyl?

From an experimental perspective, no direct oxidative addition of HBpin via complex A is observed by ¹H or ¹¹B NMR spectroscopy. In a previous study, we did observe the formation of an Fe-H/Fe-Bpin species (*Chem Commun*, **2023**, *59*, 7427), however this is only generated at elevated temperatures and when this species was subjected to a stoichiometric quantity of ketone, no reduction products were observed. From the computational calculations, the barriers to formation of **10a** from **10** (activation of B-H bond on one Fe centre) and then from **10a** to **11** (breakup of the dimer "A") are 11.1 and 11.6 kcal mol⁻¹, respectively. This step seems to be key to breakup of the dimer. Direct activation of B-H by either monomer of A is 24.1 kcal mol⁻¹ (with barrier to hydride transfer in excess of 30 kcal mol⁻¹). The cooperative transition state was extensively searched for but consistently resulted in the 10a structure we have presented, despite attempting many different starting geometries.

7.In Scheme 4, correct the labeling of intermediates 11 and 12 from "PH". Also, ensure that panels a, b, c, d in Figure 1 are properly labeled, and check the designation for compound 2i.

We thank the reviewer for highlighting this. This has now been addressed in the manuscript.

8.The mechanistic section is difficult to follow; additional calculations and clarification are needed regarding the initiation step.

We have re-written the kinetics study section and made the order in Fe and ketone easier to follow with a more detailed interpretation of the results. Both experimental and computational results

indicate initiation is not due to B-H activation, but requires a three-component transition state (as discussed in the answer to the comment "...further experimental and computational studies are needed to clarify the initiation step..." above. Additionally, our changes to the proposed catalytic cycle figure (Scheme 4, page 9) make this section fully consistent (as discussed above in our response to the comment: "My primary concern is a mechanistic inconsistency...").

9. For the monomeric Fe(I) species in Figure 3, did the DFT calculations consider all relevant spin states (doublet, quartet, sextet)? Please report the relative energies in the Supporting Information. The energies for the different spin states are: Quintet -3713.6985053877, Triplet -3713.7770516369. These are presented in the supporting information (note that the complex was neutral overall).

11. In the proposed rate-limiting transition state, the metal appears to have weak interactions with the substrates. Is this plausible? The computed barrier of 26.2 kcal/mol suggests the reaction may not occur at room temperature.

The calculated barrier is 24.2 kcal/mol with respect to the reactants, which is feasible (using the Eyring equation predicts a half-life of ~17 hours, which is in line with the experimental data, if slightly longer). All other "rate-limiting" transition states for the different pathways were considerably higher in energy, usually > 30 kcal/mol.

12. All optimized coordinates should be included in the Supporting Information.

We apologise for mistakenly omitting these, they are now included in the revised submission.

Reviewer #2 (Remarks to the Author):

The manuscript by Willcox, Robinson and coworkers reports on an iron metalloborane dinitrogen complex that catalyzed the hydroboration of ketones, cyclic esters and carbon dioxide. This is a nice piece of work that represents a detailed study on earth-abundant iron-catalyzed reactions with a range of carbonyl substrates. Both experimental and computational work has been conducted for uncovering the mechanistic aspect, along with the isolation and characterization of a novel iron-radical complex resulting from the stoichiometric reactions. While this reviewer is mostly positive on the overall quality of the work, the following factors diminish my enthusiasm about its publication in this high impact interdisciplinary journal. A more specialized journal in catalysis would be better suited. 1, both the catalyst and the catalytic reactions presented here are not new and in fact the hydroboration of carbonyl compounds has been long explored with numerous catalysts or even without a catalyst in the literature. The mechanistic study is much appreciated, yet it does not represent any conceptual advances that justify a publication in Nature Communications. The isolation of an iron-ketyl radical product was significant, but turned out not to be a key intermediate in the catalytic cycle. Thus, it did not benefit the mechanistic understanding.

We thank the reviewer for their comments. We are not suggesting that the catalyst nor the transformation itself are new, but that this class of Fe(0) complexes have rarely been screened in catalysis and therefore, offers a unique possibility to explore how these complexes behave. The mechanistic evidence (experimental and computational) supports a mechanism which hasn't been previously reported for transition metal complexes and is different to the mechanism we report in our previous study which proceeds via an Fe-H/Fe-Bpin species (*Chem. Eur. J.*, **2025**, *31*, e202501782) rather than a direct Fe(0)-mediated LLHT. Whilst the Fe(I)-ketyl species is off-cycle, this represents the first example of an Fe(I)-ketyl complex which could have further implications for other catalysis.

2, as for the catalytic reactions, the substrate scope was not well studied. More challenging substrates containing reducible functional groups should be included. It was also unfortunate to display poor catalytic activity towards general ester substrates.

We thank the reviewer for their comment regarding substrate scope. We have now addressed this issue and included more substrates which contain other reducible groups such as sulfones, esters, exo-cyclic olefins and amides (**2p-2s**). The spectroscopic data for this has been included in the supporting information. Regarding the inactivity of esters towards these conditions, we believe that highlighting the limitations of this catalyst system is important and thus demonstrates the chemoselectivity of our protocol, where we can tolerate substrates bearing esters.

Reviewer #3 (Remarks to the Author):

The article by Grose et al. describes an experimental and computational study into very interesting iron boron chemistry for hydroboration of carbonyl containing compounds, and they describe the isolation of a unique iron(I) ketyl complex in an antiferromagnetic ground state. The manuscript is well written and the topic is of high scientific interest.

Overall the work is highly interesting both in terms of experiment and theory.

I will focus my review on the computational work in this study and I ask the authors to consider the comments below:

1) In iron coordination compounds, the electronic structure may have a large flexibility in spin states depending on the detailed surroundings. This is apparent in the intricate electronic structure found for antiferromagnetically coupled complex **9**, but it will also play a role in the dinuclear N₂-bridged complex **A**. Assuming S=1 for the two Fe centres, it is not obvious how the magnetic moments of the two iron centres are coupled and what the electronic ground-state structure looks like. Further, in terms of reaction kinetics and thermodynamics the concept of two-state or multi-state reactivity involving spin crossover steps is often relevant. Therefore the electronic structure of all Fe containing species must be carefully evaluated in order to ascertain the ground-state potential energy surface.

Also: The electronic structure of species **A** should be discussed in a little detail:

- How are the two magnetic iron centres coupled?
- Why are other spin multiplicities not considered or discussed?
- How is the possibility of spin change across the PES excluded?

In Table S3, we have given the energies from the r²SCAN-3c geometry optimisations for each of the minima along the computed lowest energy reaction pathway with S=0,1,2 at each Fe centre. These results clearly indicate that S=1 is consistently the lowest energy solution. Complex **A** used in this study was previously reported by Suess and Peters in 2015 (*J. Am. Chem. Soc.* **2013**, *135*, 4938–4941) in which they claim there is an antiferromagnetic coupling between the two S = 1 iron centres, with the two pseudotetrahedral Fe centres each having different local geometries in the solid state.

2) With regard to reactivity the entropic contributions cannot be neglected in my opinion. This is particularly true for reaction steps involving the cleavage/formation of dimeric structures or, in general, steps involving change in particle numbers.

In Figure 3, the updated text (with slightly updated values, where thermal enthalpic contributions have also been included) now indicates that these are Gibbs free energy values, where the entropy contribution is from translational entropy. An alternative free energy profile is given in the Supporting Information (Figure S8), where vibrational and rotational entropy have also been included. In this case, some of the early transition states' energy barriers are raised, but the rate-limiting step stays close to the same position (relative to **A**).

3) I would be interested to learn more about the authors' perspective on why the iron ketyl radical 9 exhibits such unusual antiferromagnetic features. Are there particular properties of the iron-boron interaction and/or the iron-carbonyl bond that govern the antiferromagnetic ground state? Or conversely, what inhibits this antiferromagnetic behaviour in other iron-ketyl compounds? Further elaboration would enhance the impact of the manuscript.

It would appear that the steric bulk of the group attached to the boron-containing ligand, and the fact that there are several phenyl groups in close proximity, allows the open-shell "singlet" behaviour (of the antiferromagnetic coupled electrons, in addition to the two alpha spin electrons which give the overall $S=1$ spin state). While we can note that this exists with DFT, it would require further investigation with a multireference method (e.g. CASSCF/CASPT2). However, this would be challenging given the number of interacting orbitals. As this was an off-cycle product rather than central to the catalytic process, we have (for now) decided not to investigate in further detail.

Further specific remarks

Main text:

Computational study of 9:

How does the computed molecular structure/geometry compare with that extracted from XRD?

The geometry of 9 was optimised directly from the XRD structure and has an RMSD of 0.34 Å. The overall structure matches qualitatively, with very minor differences in atomic positions, presumably as a result of gas phase / condensed phase (calculations) vs. packing effects in the crystal structure.

Please report the relative energies of the neighbouring (pure) spin multiplicities, $S=2$ quintet and $S=0$ closed-shell singlet.

These are now reported in Table S3 in the Supporting Information.

Please state the isodensity value for the spin density plot in Figure 2d.

The figure caption has now been updated to include the isodensity value (which is 0.02 \AA^{-3}).

Please state the $\langle S^2 \rangle$ value for 9 also somewhere in the main text (e.g. in the caption of Figure 2d).

The $\langle S^2 \rangle$ value of 3.28 is now reported in the figure caption as recommended.

The spin populations (NPA or Mulliken) on the relevant atomic sites (Fe, B, Fe-coordinated phenyl, O, rest of the ketyl group) should be reported - in the ESI as appropriate.

These quantities are now reported in the Supporting Information. Three atoms have substantial spin density: Fe, B and the C (of the $C=O$ group coordinated to Fe).

Energy profile for hydroboration:

It is not clear which energy values are reported for the computed potential energy surface:

On the y-axis of Figure 3 " $\Delta H + ZPE$ " is given. This is a conflicting description. ΔH would already include ZPE. Perhaps " $\Delta E + ZPE$ " is meant? Then these values would correspond to zero K data. Is this the case here?

We apologise for the confusing (and technically incorrect) labelling given here. These are Gibbs free energy values, where the entropic term is taken from the translational entropy (as discussed above).

Please report Gibbs energies including entropy contributions at r.t. instead of just enthalpies (or energies at zero K). This is particularly important for comparison with experiment and for assessment whether a barrier height is rate limiting or not.

Please see comment above.

If I understand correctly, only the S=1 triplet spin surface is considered. Other spin multiplicities might play a role here, and these should be investigated and discussed in the same detail (or excluded if applicable).

The energies for the optimised geometries with different spin multiplicities are now given in the Supporting Information (Table S3) with a sentence added to page 8 of the manuscript. In all cases, the S=1 per Fe centre is the lowest in energy,

Supporting information:

Please report the corresponding transition wavelengths/energies of the TD-DFT results. How does the computed spectrum compare to the experimentally observed one?

The computed transition wavelengths are now given in the Supporting Information. The calculated wavelengths are 490 nm (c.f. expt. 522 nm) and 605 nm (c.f. expt. 652 nm). These are in reasonable agreement, given that explicit solvent hasn't been used in the calculations.

Figures S8-S10:

It is not clear which spin surface the PES refers to. Please report specifics on the electronic structure, at least the spin multiplicity of the species in the PES.

The Figure captions have been updated to reflect the S=1 spin multiplicity of the different calculations.

For comparison with experiment, Gibbs free energies at r.t. should be reported instead of "enthalpies plus ZPE".

Please see comment(s) above.

Please carefully check and amend the References in the ESI.

Here are two examples:

Reference 13 seems incomplete.

The authors' list in Reference 15 for the QChem package doesn't look right.

Reference 13 has been updated with an access date. Reference 15 has been updated with the first author's name; however, as there are 220 authors on this paper, we have truncated the list using standard notation (i.e. "et al.").

Please include Cartesian coordinates, total energies and spin expectation values for all computed species.

The Cartesian coordinates have now been uploaded to the Supporting Information – as we responded to Reviewer 1 above, we apologise for accidentally omitting these. Total energies (from the ω B97X-D3/def2-TZVP calculations) and corresponding $\langle S^2 \rangle$ values are now given in Table S4 of the Supporting Information.

The authors attempted to revise the manuscript in accordance with the reviewers' comments; however, the revisions and responses remain unsatisfactory.

1. The monomeric Fe(I) species can exist as a doublet, quartet, or sextet. However, the authors calculated only the quintet and triplet states. What is the total charge of this complex? Did the authors also calculate the Fe(II) species? If not, this omission raises significant concerns about the analysis.

From the reviewer's original comments (and that above), we're not sure if they're referring to the ketyl radical (**9**, shown in Figure 2), or structures in the reaction profile (e.g. **11**, **13**, Figure 3 and Scheme 4). For the ketyl radical species (**9**), the SQUID data is suggestive of an antiferromagnetically coupled pair of electrons (one from iron and one from the ketyl fragment) due to the curves not plateauing at the higher temperature regime, and two unpaired electrons on the iron centre ($S = 1$, μ_{eff} of 2.78). If the Fe is in the +1 oxidation state i.e. d^7 centre, with an antiferromagnetic coupling to the ketyl radical, this will result in an $S = 1$ ground state. For an Fe(II) system (d^6), with an antiferromagnetic coupling, then a Krämers doublet is possible, however this would result in the observation of a signal in the EPR spectrum. When the EPR was performed at X-, L- and K- band, both frozen and solution, no signals were observed, suggesting that ketyl radical species **9** has a whole integer spin rather than a non-integer spin.

This would also be true for **11** and **13** as well. The overall charge (in each case) is neutral, such that the total number of electrons is even, hence doublets, quartets etc are not possible, in line with the experimental evidence (SQUID + EPR). We have given energies calculated for the different spin states of the reaction profile (singlets, triplets and quintets) in Table S3 of the Supporting Information.

2. According to the energy span model, the total barrier is calculated to be 26.2 kcal/mol, as noted in well-established literature within the field of computational catalysis (Accounts of Chemical Research, 2011, 44, 2, 101-110). There should be a more favorable reaction pathway that can satisfactorily account for the experimental observations. If the reaction occurs at room temperature, the barrier should ideally be below 25 kcal/mol.

The values presented in Figure 3 are all relative to **A** and are calculated using $\Delta E + \text{ZPE} + \text{translational entropy correction}$, to give an approximate ΔG value (this is described in Figure 3 and the computational details in the Supporting Information). In Figure S8 (Supporting Information), we present the reaction profile using all enthalpy and entropy corrections from the implicit solvation model in the calculation of ΔG , leading to a barrier height of 18.3 kcal mol⁻¹ (consistent with the reviewer's suggestion of the energy span model). The other pathways considered all have barrier heights in excess of 30 kcal mol⁻¹, irrespective of which definition of ΔG is used.

3. The proposed mechanism suggests a concerted transition state involving a four-membered ring. Notably, there is no indication of coordination or bonding interaction between the metal center and the substrate in this transition state. What is the actual role of the catalyst in this scenario? This appears to be an unprecedented form of catalysis. The authors may well achieve a similar barrier without the catalyst.

The mechanism we have proposed shows a stepwise formation of the product: in the first transition state, the B(HBpin) – O(ketone) bond is formed, followed by the second transition state in which hydride transfer from the Bpin to carbon occurs. The Fe centre plays a role in formation of the initial B-O bond, by increasing the Lewis acidity of the boron centre upon coordination of HBpin (**11**), allowing the ketone to react to form the intermediate (labelled as ts_11-13_int.xyz in the supporting data) which goes to the rate determining transition state. In this transition state, steric attraction supports the hydride transfer resulting in a lowering of the energy barrier for this process (relative to the uncatalysed reaction). In the manuscript, this is described on page 8 (first paragraph of “DFT studies”), which notes that the uncatalysed reaction is ~ 5 kcal mol⁻¹ higher than the catalysed reaction.

From an experimental viewpoint, no reaction occurs in the absence of the Fe(0) species over the 5-hour time frame for this substrate. Based on our observations, we believe the role of the catalyst is to weaken the B–H bond to facilitate the hydride transfer.

In the DFT section of the manuscript, we have broken the text up so that it is not suggestive of a concerted pathway for the transfer of the boron to oxygen and the hydride addition step.